# Metabolic Biomarkers Affecting Cell Proliferation and Prognosis in Polycythemia Vera

**DOI:** 10.3390/cancers14194913

**Published:** 2022-10-07

**Authors:** Ziqing Wang, Yan Lv, Erpeng Yang, Yujin Li, Dehao Wang, Guang Hu, Yumeng Li, Mingjing Wang, Weiyi Liu, Mingqian Sun, Xiaomei Hu

**Affiliations:** 1Department of Hematology, Xiyuan Hospital, China Academy of Chinese Medical Sciences, Beijing 100091, China; 2Graduate School, Beijing University of Chinese Medicine, Beijing 100029, China; 3Graduate School, China Academy of Chinese Medical Sciences, Beijing 100700, China; 4Institute of Basic Medical Sciences, Xiyuan Hospital, China Academy of Chinese Medical Sciences, and Beijing Key Laboratory of Pharmacology of Traditional Chinese Medicine, Beijing 100091, China

**Keywords:** polycythemia vera, cell proliferation, prognosis, metabolic biomarkers, metabolomics

## Abstract

**Simple Summary:**

Polycythemia vera (PV) is a malignant neoplastic disease. Abnormal cell metabolism is a new feature of malignant proliferation of tumor cells. Our study described the global metabolic profile of PV patients, analyzed their relationship with cell proliferation, screened prognosis-related metabolic biomarkers. Notably, fatty acid metabolism, glucose metabolism, sphingolipid metabolism, and amino acid metabolism were significantly altered in PV. In addition, Cer(d18:2/22:6-2OH(7S, 17S)) and SM(d18:0/PGF1α) were closely associated with JAK2 mutations, which may contribute to the proliferation of peripheral blood cells in PV patients. The elevated levels of four potential biomarkers may provide a reference for poor prognosis in PV patients.

**Abstract:**

Polycythemia vera (PV) is a malignant clonal hematological disease of hematopoietic stem cells characterized by the proliferation of peripheral blood cells, and JAK2 mutation is one of the main causes of PV peripheral blood cell proliferation. Abnormal cell metabolism is a new feature of malignant proliferation of tumor cells, but the role of metabolism in the pathogenesis and prognosis of PV remains unclear. We analyzed metabolic differences of peripheral blood sera between 32 PV patients and 20 healthy controls (HCs) by liquid chromatography–mass spectrometry (LC–MS) to investigate their relationship with cell proliferation and to screen for prognosis-related metabolic biomarkers. Compared to HC, 33 endogenous metabolites were significantly changed in PV and were involved in fatty acid metabolism, glucose metabolism, sphingolipid metabolism, and amino acid metabolism pathways. Among them, seven metabolites were closely associated with JAK2 mutations, 2 of which may contribute to the proliferation of peripheral blood cells in PV patients. A set of potential prognostic metabolic biomarkers containing four metabolites was identified by a receiver operating characteristic (ROC) curve according to the risk stratification of the PV patients and their combined AUC value of 0.952, with a sensitivity of 90.905% and specificity of 90.909% at the optimal cutoff point. Metabonomics is an important tool for the study of the pathogenesis of PV and the relationship between JAK2 gene mutation. Furthermore, the potential biomarkers of this study may provide a reference for the prognosis of PV.

## 1. Introduction

Polycythemia vera (PV) is a malignant clonal myeloproliferative neoplasm (MPN) of hematopoietic stem cells characterized by polycythemia with varying degrees of myeloid granulocyte and megakaryocyte proliferation [1]. The incidence rate of PV in the US Surveillance, Epidemiology, and End Results database is 1.09/100,000, the median age of onset is approximately 65 years old [2], and the median survival time is approximately 15 years. PV has a hidden onset and slow progression, often accompanied by hepatosplenomegaly, thrombosis, and bleeding tendency, and some patients may transform into myelofibrosis (MF) and acute myeloid leukemia (AML) in late stages. In 2005, Baxter et al. [3] first found that the rate of JAK2 gene mutations in BCR/ABL negative MPN was up to more than 50%, and the rate of JAK2 mutations in PV patients was greater than 90%, which was strongly associated with the occurrence and development of MPN. In PV, mutations in the driver gene JAK2 can lead to sustained activation of JAK-STAT and its downstream pathways, which participate in regulating the proliferation and differentiation of hematopoietic cells [4].

Abnormal cell metabolism is a new feature of the malignant proliferation of tumor cells discovered in recent years, it participates in almost all physiological processes of tumor cells including proliferation, metastasis, and apoptosis [5]. With the maturation of research mechanisms, some shared metabolic features of tumors have been gradually recognized. Enhanced glycolysis allows tumor cells to adapt to the anaerobic microenvironment [6], and abnormal lipid metabolism provides energy and material basis for tumor cells [7] and includes enhanced fatty acid oxidation, disturbed sphingolipid metabolism, increased amino acid utilization, and a suppressed tricarboxylic acid cycle [8,9,10].

PV, as one of the BCR/ABL-negative MPNs, is a malignant neoplastic disease, and JAK2 mutation is one of the exact factors in its pathogenesis. In recent years, some scholars have reported abnormal metabolism in MPN. Rao et al. [11] found in a JAK2 mutant MPN mouse model that enhanced JAK2 mutation disrupted the metabolic homeostasis of MPN cells, resulting in systemic metabolic changes in vivo, including elevated levels of glycolysis, oxidative phosphorylation, adipose tissue atrophy, and early death. Zhan et al. [12] found increased glutamine metabolism and upregulation of glutaminase in both JAK2V617F mutant cells and clonally derived erythroid progenitor cells from JAK2V617F-positive MPN patients, as well as increased glutaminase levels with disease progression. Preliminary studies on the metabolic characteristics of MPN patients have been conducted using metabolomics. Forte et al. [13] observed disturbances in the metabolism of the endogenous cannabinoid system in MPN patients using liquid chromatography-mass spectrometry (LC-MS) and found that arachidonoyl ethanolamide levels in MPN patients were positively correlated with platelet counts.

With the recent development of metabolomics technology, an increasing number of metabolism-related biomarkers have been used to evaluate the prognosis of malignant tumors. Two plasma biomarkers, succinic acid, and gluconic acid have been identified to effectively diagnose the progression and metastasis of pancreatic cancer, providing a reference for clinicians to determine prognosis and treatment options [14]. Another study revealed distinct metabolic phenotypes that distinguish low-grade from high-grade colon cancer and demonstrated that metabolomic phenotyping is a potentially important molecular pathology for the diagnosis and prognosis of solid tumors [15].

In this study, we investigated the metabolic profile of PV patients using untargeted LC-MS and analyzed the metabolites associated with JAK2 mutations and their relationship with blood cell counts, which can help us better understand the pathogenesis associated with PV metabolism and explore its intrinsic connection with JAK2 mutations and cell proliferation. In addition, metabolites associated with the prognosis of PV patients were screened by risk stratification, which provides a reference for assessing patient prognosis.

## 2. Materials and Methods

### 2.1. Patient Inclusion and Sample Collection

This study was approved by the Ethics Committee of Xiyuan Hospital, China Academy of Chinese Medical Sciences in accordance with the guidelines in the Declaration of Helsinki (2019XLA024-3). All subjects signed a written informed consent form before the start of this study. Between September 2020 and January 2022, 32 PV patients (24 cases in the JAK2 mutation group, including 22 cases with JAK2 V617F mutation, two cases with JAK2 exon12 mutation, and eight cases in the JAK2 unmutated group) were recruited from Xiyuan Hospital of China Academy of Chinese Medical Sciences to participate in this study according to a JAK2 mutated to unmutated ratio of 3:1, and 20 healthy volunteers were recruited as the control group. Among the PV patients, there were 23 males (71.9%) and nine females (28.1%), with a median age of 54.5 (21~77) years. All patients met the World Health Organization (WHO) (2016) diagnostic criteria for PV [16]. Sex- and age-matched control serum samples were collected from healthy volunteers without metabolic diseases (mainly including diabetes mellitus, metabolic syndrome, and thyroid disease) who were examined at our hospital, based on medical examination reports within one year. Among the healthy volunteers, there were 11 males (55%) and nine females (45%), with a median age of 49.5 (28~78) years. The clinical characteristics of PV patients are listed in Appendix A.

After fasting overnight, venous blood was collected from patients and healthy volunteers between 8 and 9 a.m. and put into vacutainer tubes containing inert separating gel and clot activator (Greiner Bio-One GmbH, Frickenhausen, Germany). After standing at room temperature for 30 min, the collected samples were centrifuged for 20 min at 4 °C and 1000 g, and the supernatant was collected and stored in aliquots at −80 °C.

### 2.2. Prognostic Risk Stratification

The prognostic point system proposed by Tefferi et al. [17] was used to classify PV patients into low-risk (0 points), intermediate-risk (1 or 2 points), and high-risk groups (≥3 points) according to age (5 points for ≥67 years and 2 points for 57–66 years), WBC > 15 × 109/L (1 point) and venous thrombosis (1 point). 

### 2.3. Detection Method of JAK2 Mutations

Mutation analysis of DNA from peripheral blood using next-generation sequencing technology. Five ml of venous blood was drawn in the early morning under fasting conditions, peripheral blood single nuclei cells were extracted and whole genomic DNA specimens were prepared. The DNA quality control qualified samples were fragmented for whole genome and exon library construction and quality control. The quality control qualified samples were subjected to Illumina HiSeq2500 high-throughput sequencing. The filtered data were compared to the human reference genome HG19 using Burrows-Wheeler alignment and quality control of the corresponding indicators, and the output data were counted. Single nucleotide variants (SNPs), small fragment insertion-deletion variants (InDel), and mutation hotspots were detected using GATK to annotate and count variants in genes such as JAK2. The tumor mutation burden (TMB) generally refers to the number of somatic nonsynonymous mutations per megabase pair in a given genomic region. TMB was calculated as follows: TMB = somatic/L, where somatic represents the number of somatic variants detected with nonsynonymous mutations; L represents the effective coverage area (located at the intersection of a 50 bp extension upstream and downstream of the full exon capture interval region and a 2 bp extension upstream and downstream of the CDS region).

### 2.4. Materials and Reagents

Acetonitrile and methanol were Optima™ LC-MS grade and obtained from Fisher Scientific (Fair Lawn, NJ, USA). Formic acid was purchased from J. T. Baker (Philipsburg, The Netherlands). 

### 2.5. Experimental Instruments

The experimental instruments used in this study included the following: Waters ACQUITY UPLC liquid chromatography system, Q-TOF SYNAPT G2 HDMS mass spectrometer, and MassLynx v4.1 chromatographic workstation (Waters, Milford, MA, USA); a high-speed refrigerated centrifuge (Eppendorf, Hamburg, Germany); Vortex Genius 3 vortex oscillator (IKA company, Staufen, Germany); and the Milli-Q Gradient A10 water purification system (Millipore, Billerica, MA, USA).

### 2.6. Serum Sample Preparation

The serum samples were thawed at room temperature before analysis. A volume of 200 µL acetonitrile was added to 50 µL serum and shaken vigorously. Then, the mixture was stored at room temperature for 10 min and centrifuged at 16,090× *g* for 10 min at 4 °C. The supernatant was analyzed by LC-MS.

### 2.7. LC-MS Analysis

Liquid chromatography was performed using an LC instrument (ACQUITY UPLC, Waters, Milford, MA, USA) with a HILIC column (BEH, Amide, 1.7 µm, 2.1 × 100 mm, Waters, Wexford, Ireland). For chromatography analysis, the mobile phase consisted of water containing 0.1% formic acid (A) and acetonitrile containing 0.1% formic acid (B). A gradient program was used as follows: 30% B; 0~10 min, 30–90% B; 10~11 min, 90 B%; 11~13 min, 90~30% B; stop 15 min. The column was operated at 40 °C, the injection volume was 3 µL, and the flow rate was 0.3 mL/min.

Mass spectrometry was performed using a Waters Xevo G2-XS series Time of Flight mass spectrometer equipped with an electrospray ionization (ESI) interface. High purity nitrogen (N2) was used as the sheath and auxiliary gas, and high purity argon was used as the collision gas. The ESI-MS spectra were respectively acquired in positive and negative ion modes; positive ion mode, capillary voltage +3 KV; negative ion mode, capillary voltage −2.5 kV. The mass scan was from m/z 50 to 1200, and the data were recorded in the centroid. Meanwhile, the fragment information was collected, and the scanning time was 0.2 S. The standard curve of the mass axis was established by using a sodium formate standard. In the process of data acquisition, the lockspray correction system was used for real-time quality correction of leucine enkephalin (LE, [M + H] + = 556.2771, [M−H] + = 554.2615).

### 2.8. Statistical Analysis

The original mass spectrometry data were transferred to Progenesis QI (Waters, UK) software. Several steps, such as peak alignment, peak extraction, and peak identification, were carried out to obtain the metabolite information table, including the mass charge ratio, retention time, and ion area. The metabolites were identified by the biochemical database HMDB (http://www.hmdb.ca/, accessed on 17 July 2022). The data were imported into EZinfo 2.0, and the metabolites with a fold change (FC) value > 2 were screened out, with *t*-test *p* < 0.05 as the limited condition for screening potential biomarkers. For multiple linear regression analysis, the results were introduced into MetaboAnalyst 4.0 (http://www.metaboanalyst.ca/MetaboAnalyst/, accessed on 20 July 2022), and partial least squares discriminant analysis (PLS-DA) was used to identify the differences between groups. The quality of the model was evaluated by the model parameters Q^2^ (predictability of the model) and R^2^ (fit of the model). Metabolic pathway analysis was also performed by MetaboAnalyst (https://www.metaboanalyst.ca/, accessed on 20 July 2022).

In addition, the Mann–Whitney U test was performed using SPSS 23 (IBM Corporation, Armonk, NY, USA) to assess the statistical significance of the different groups. Spearman’s correlation test was used to analyze the correlation of metabolites with clinical indicators, and the results were visualized by heatmap using R 3.6.1. (Microsoft, Redmond, WSU, USA) Binary logistic regression analysis and receiver operating characteristic (ROC) curve analysis were performed to identify prognostic biomarkers.

## 3. Results

### 3.1. Analysis of the Metabolic Profiles of PV Patients and Healthy Controls

To assess whether there were significant changes in the metabolic profiles of PV patients and healthy controls, we performed a multifactorial analysis of the metabolic profiles of the two groups. Before analyzing the samples, qualitative control (QC) samples were used to investigate the instrument precision, method precision, and sample stability. In this study, a QC sample was created from equal volume subaliquots of all samples and injected every ten samples to monitor the stability of the analysis in the run sequence. The data collection of analytical samples can be conducted only when the methodology is qualified. Ten chromatographic peaks were randomly selected, and the RSD values of peak area and retention time were calculated. The results showed that the RSD of the peak area and retention time of positive ion and negative ion chromatography were less than 20% and 1.0%, respectively, which proved that the methodological experiment was qualified. Figure 1 shows the score plots of the PLS-DA in positive and negative modes. For the data collected in HILIC mode, the results of the PLS-DA showed that the PV groups could be well distinguished from the healthy control (HC) group, indicating that more significant metabolic changes occurred in the serum of PV patients. The quality of the model was good, with variance explained R^2^ = 0. 965 and variance predicted Q^2^ = 0. 843. For the PLS-DA of the negative mode (R^2^ = 0.955, Q^2^ = 0.801), clear separations between the PV and healthy groups were also observed. Data from both the positive and negative modes suggested that the serum metabolites of PV patients were significantly changed.

Subsequently, we identified endogenous metabolites between PV patients and healthy controls. According to the variable importance in projection (VIP) score and FC analysis of the PLS-DA model, combined with Student’s *t*-test, the differential metabolites satisfying VIP > 1, FC > 2, and *p* < 0.05 were screened. A total of 33 endogenous metabolites were initially identified as potential biomarkers for PV diagnosis under positive and negative ion modes, which were imported into the database for searching and confirmation. These potential biomarkers were eventually identified as fatty acids, acylcarnitine, sphingolipids, amino acids, etc., and the identification results are shown in Table 1. The screened biomarkers were subjected to hierarchical clustering analysis to obtain a heatmap (Appendix A), which showed that the PV and HC groups could be distinguished and clustered well within the group, indicating that the screened biomarkers were more reliable.

In order to clarify the abnormal metabolic pathways in PV, we performed an enrichment analysis of the metabolic pathways of the above biomarkers. The screened potential biomarkers were imported into MetaboAnalyst 5.0 for metabolic pathway analysis. It is generally accepted that changes occurring at key locations in the network have a serious impact on the occurrence of events, and pathways with impact values >0.1 are considered potential target metabolic pathways. The results showed that there were nine important metabolic pathways associated with PV, namely: D-glutamine and D-glutamate metabolism, sphingolipid metabolism, pyruvate metabolism, alanine, aspartate and glutamate metabolism, arginine and proline metabolism, tryptophan metabolism, lysine degradation, arginine biosynthesis and glycolysis/gluconeogenesis (Figure 2). These can be mainly attributed to fatty acid metabolism, glucose metabolism, sphingolipid metabolism, and amino acid metabolism.

### 3.2. Effect of JAK2-Associated Metabolic Abnormalities on Cell Proliferation in PV Patients

Mutations in JAK2 can activate JAK-STAT and its downstream pathway to cause blood cell proliferation. To clarify the relationship between JAK2 and peripheral blood cell counts, we analyzed the blood cell counts of JAK2-mutated and unmutated patients. Platelet count (PLT) and hematocrit (HCT) levels were significantly higher in JAK2-mutated PV patients than in JAK2-unmutated PV patients (*p* = 0.006; *p* = 0.044; Figure 3A,B). Further analysis of the correlation between blood cell counts and JAK2 mutational burden revealed a significant positive correlation between JAK2 mutational burden and white blood cell count (WBC) and PLT levels (*r* = 0.62, *p* = 0.001; *r* = 0.55, *p* = 0.007; Figure 3C,D). We found a significant positive correlation between JAK2 mutational burden and WBC levels, however, there was no difference in WBC levels observed between the JAK2 mutated and unmutated groups. Therefore, we further divided PV patients with JAK2 mutation into high burden group (mutational burden ≥ 50%, *n* = 16) and low burden group (mutational burden < 50%, *n* = 8), and the comparison revealed that WBC levels were significantly higher in the JAK2 high burden group than in the low burden group and the unmutated group (*p* = 0.01; *p* = 0.014; Figure 3E).

To screen for endogenous metabolites associated with JAK2, PV patients were divided into a JAK2-mutated group (*n* = 24) and a JAK2-unmutated group (*n* = 8) according to their JAK2 mutation status, and the PLS-DA model was used to represent the changes in metabolic profiles of patients in the JAK2-mutated and JAK2-unmutated groups. Good separation between the two groups of samples was observed in the PLS-DA score plot in both positive and negative ion modes (Figure 4A,B), and the model had good fitness and predictive power (R^2^ = 0.954, Q^2^ = 0.633 in positive ion mode; R^2^ = 0.996, Q^2^ = 0.612 in negative ion mode), indicating a significant difference in the metabolic profiles of patients in the JAK2-mutated and unmutated groups. Further analysis of 33 endogenous metabolites identified seven metabolites that differed significantly between the JAK2 mutated and unmutated groups. The levels of Cer(d18:2/22:6-2OH(7S, 17S)), SM(d18:0/PGF1α), CerP(d18:1/16:0), glutamic acid, lactic acid, melibiose and xylose were significantly higher in the JAK2-mutated group than in the JAK2-unmutated group (*p* = 0.001, *p* < 0.001, *p* = 0.011, *p* = 0.017, *p* = 0.021, *p* = 0.022 and *p* = 0.019, respectively; Figure 4C).

To clarify the relationship between metabolic abnormalities and the JAK2 mutational burden, we further analyzed the relationship between the above seven endogenous metabolites and the JAK2 mutational burden. We observed a positive correlation between two endogenous metabolites and the JAK2 mutational burden: Cer(d18:2/22:6-2OH(7S, 17S)) (*r* = 0.817; *p* < 0.001; Figure 5A) and SM(d18:0/PGF1α) (*r* = 0.463; *p* = 0.021; Figure 5B).

Our study above found that peripheral blood cells were significantly elevated in JAK2-mutated PV patients compared to JAK2-unmutated PV patients and positively correlated with JAK2 mutational burden. Moreover, Cer(d18:2/22:6-2OH(7S, 17S)) and SM(d18:0/PGF1α) levels differed between JAK2-mutated and unmutated patients and were positively correlated with JAK2 mutational burden. Thus, we evaluated the relationship between these two endogenous metabolites and blood cell counts. Here, we found that Cer(d18:2/22:6-2OH(7S, 17S)) levels were positively correlated with WBC (*r* = 0.650, *p* < 0.001; Figure 6A ), and SM(d18:0/PGF1α) levels were positively correlated with WBC and PLT (*r* = 0.444, *p* = 0.011 and *r* = 0.623; *p* < 0.001, respectively; Figure 6B,C). In addition, we analyzed the correlation between 33 endogenous metabolites and WBC, HGB, PLT, and HCT at diagnosis in patients with PV, and the correlation heatmap is shown in Appendix A.

### 3.3. Potential Metabolic Biomarkers Relevant to the Prognosis of PV Patients

According to the prognostic point system proposed by Tefferi et al. [17], PV patients can be divided into low-risk group (*n* = 11) and intermediate/high-risk group (*n* = 21). Further analysis of 33 endogenous metabolites showed that compared to the low-risk group, the levels of Cer(d18:2/22:6-2OH(7S, 17S)), SM(d18:0/PGF1α), CerP(d18:1/16:0), octadec-13-enoylcarnitine, glutamic acid, lactic acid, melibiose and 1-pyrroline-5-carboxylic acid in the intermediate/high-risk group were significantly elevated (*p* = 0.001, *p* = 0.001, *p* = 0.006, *p* = 0.013, *p* = 0.025, *p* = 0.016, *p* = 0.014 and *p* = 0.016, respectively; Figure 7A).

We next tested the diagnostic potential of eight biomarkers by ROC curve analysis. Eight biomarkers had AUC values ranging from 0.745 to 0.87 and all showed moderate efficiency (AUC = 0.70–0.90). In addition, we used binary logistic regression to analyze the combined biomarkers and then determine the ROC curve. A set of potential biomarkers containing four endogenous metabolites with an AUC > 0.77 was found to provide an AUC of 0.952 (Figure 7B), with a sensitivity of 90.905% and specificity of 90.909% at the optimal cutoff point. Therefore, elevated levels of Cer(d18:2/22:6-2OH(7S, 17S)), SM(d18:0/PGF1α), CerP(d18:1/16:0), and octadec-13-enoylcarnitine could be used as potential metabolic biomarkers of poor prognosis in patients with PV.

## 4. Discussion

PV is a hematological malignancy that occurs with an excessive clonal proliferation of myeloid cells in hematopoietic stem or progenitor cells [1]. Abnormal cell metabolism is one of the ten metabolic characteristics of tumors. As a hematological malignancy, the role of metabolism in PV is also worthy of attention. In recent years, preliminary studies on metabolic abnormalities in MPN have been conducted, and abnormalities in glucose metabolism, glutamine metabolism, and lipid metabolism have been identified in MPN, which may be involved in the pathogenesis and progression of MPN [11,12,13]. Here, through the global metabolomic analysis of the PV and HC groups, we found significant metabolic disturbances in PV patients. Thirty-three potential biomarkers were further screened, most of which were fatty acids, acylcarnitines, sphingolipids, and amino acids, mainly involved in fatty acid metabolism, glucose metabolism, sphingolipid metabolism, and amino acid metabolism.

Among these endogenous metabolites, the levels of five acylcarnitines and four fatty acids were significantly higher in the PV group than in the HC group. The results indicated that fatty acid metabolism and the carnitine shuttle system were disrupted in the PV group. Fatty acids are activated by forming fatty acyl-CoAs, which are subsequently imported into the mitochondrial matrix via the carnitine shuttle system for beta-oxidation [18,19]. The disruption of carnitine metabolism leads to mitochondrial dysfunction, which may play a role in the energy metabolism of PV. Furthermore, four fatty acids were increased in the PV groups, which indicated the systemic disruption of fatty acid beta-oxidation. The active enzymes involved in the mitochondrial intermediary supplier pathway may be disrupted, which may cause the abnormal metabolism of carnitines and fatty acids, and the accumulation of acylcarnitine further leads to the accumulation of fatty acids [20]. In conclusion, there are systemic changes in energy metabolism in PV patients, including glucose metabolism and fatty acid metabolism.

Glucose metabolic reprogramming is the most representative metabolic phenotype in tumors, and the Warburg effect (aerobic glycolysis) is the most significant phenotype in which glucose metabolic reprogramming occurs in tumors [6]. Tumor cells produce large amounts of lactic acid through the aerobic glycolysis pathway, causing lactic acid accumulation while acquiring energy and a microenvironment conducive to their abnormal growth, proliferation, and metastasis [21]. Meanwhile, glycolysis is closely related to pyruvic acid metabolism. During glycolysis, after glucose is converted to pyruvic acid, it no longer enters the TCA cycle but synthesizes lactic acid in the cytoplasmic matrix and is induced by PDHK1 to increase the rate of conversion of pyruvic acid to lactic acid, thus promoting tumor development [22]. Previously, enhanced glycolysis was observed in MPN and has been shown to be associated with JAK2 mutations [11,23,24]. Our study found disturbances in glycolytic and pyruvate metabolic pathways in the serum of PV patients, including elevated levels of glucose, glutamic acid, and lactic acid and decreased levels of pyruvic acid and lactoylglutathione, which may be related to the enhanced conversion of pyruvic acid to lactic acid. These metabolic disturbances, consistent with the findings of Rao and Li [11,23], suggest active glycolysis in PV, indicating that PV pathogenesis may be related to abnormal energy metabolism.

Sphingolipids are the major components of cell membranes, and the major bioactive sphingolipids include ceramide, sphingosine, sphingomyelin, and sphingosine 1-phosphate [25]. Sphingolipid metabolism is involved in biological processes such as apoptosis, autophagy, necroptosis, the inflammatory microenvironment, endoplasmic reticulum stress, and cell cycle arrest in cancer cells [26]. In this study, compared with controls, we found that many sphingolipids, including sphingosine, phytosphingosine, ceramides, and sphingomyelin, showed obvious changes in the PV group. Ceramide plays an important role in the metabolism of sphingolipids, which can be formed by the hydrolysis of sphingomyelin by sphingolinases. Ceramide can activate several important pathways for the induction of apoptosis, which is closely related to the occurrence and development of tumor disease [27]. The sphingolipid metabolism of PV has not been reported, but abnormal sphingolipid metabolism has been observed in other myeloid malignancies. Similarly, upregulation of sphingomyelin, sphingosine, and ceramide has been observed in MDS or AML [28,29], suggesting that impaired sphingolipid metabolism may play a role in MDS and AML.

Amino acids are the basic component of proteins and are important raw materials for cellular anabolism, with essential functions in redox homeostasis, energy regulation, biosynthetic support and maintenance of endostasis, and amino acid metabolism plays an important role in maintaining the growth and proliferation of tumors [30]. In this study, compared with HC, some amino acids were disrupted in PV, including glutamic acid, tryptophan, and aminoadipic acid, which are mainly involved in the metabolic pathways of glutamine and glutamate metabolism, arginine biosynthesis, and tryptophan metabolism. Wang et al. [31] similarly observed metabolic disturbances of multiple amino acid metabolic pathways, glutamine and glutamate metabolism, and tryptophan metabolism in AML. Tumor cells regulate glutamate metabolism to provide precursor materials for their rapid growth and proliferation [32]. It has been shown that the glutamate/cystine antiporter SLC7A11 can enhance the glucose dependence of renal cancer cell lines and mesothelioma cell lines by exporting glutamate [33], and tryptophan metabolism confer enhanced proliferation and metastasis of cancer cells by regulating immune cell function [34]. Therefore, changes in glutamate and tryptophan may be potential biomarkers for PV.

JAK2 is one of the driving genes of PV, and the mutation rate in PV can be as high as 80–90% [35,36]. The latest guidelines now include JAK2 mutations as one of the main diagnostic criteria for PV [16]. JAK2 mutation can lead to sustained activation of the JAK-STAT pathway and its downstream pathways, regulating the proliferation and differentiation of hematopoietic cells [37]. In this study, we found that the levels of Cer(d18:2/22:6-2OH(7S, 17S)) and SM(d18:0/PGF1α) in the JAK2-mutated PV patients all increased significantly and showed a significant correlation with the JAK2 mutational burden and peripheral blood cell counts, suggesting that these two metabolites may promote the proliferation of peripheral blood cells in PV patients under the regulation of the JAK-STAT pathway. The regulatory roles between ceramide and sphingomyelin and the JAK-STAT pathway have been reported in the literature, providing a basis for this speculation; however, the specific regulatory mechanisms between them and the JAK-STAT pathway have not been well defined. It has been reported that ceramide can induce the activation of the JAK-STAT pathway in human hepatocellular carcinoma HepG2 cells, microglia and human fibroblasts [38,39,40]. It has also been demonstrated in B16 melanoma cells that the JAK-STAT pathway regulates ceramide levels [41]. Additionally, Komuro et al. [42] found that alterations in sphingomyelin affected epidermal STAT3 phosphorylation in dermatitis mice and that sphingomyelin slowed cell proliferation by inhibiting STAT3. In summary, Cer(d18:2/22:6-2OH(7S, 17S)) and SM(d18:0/PGF1α) may be involved in the proliferation of peripheral blood cells in PV patients through interaction with the JAK-STAT pathway. Moreover, these two metabolites are sphingolipids, suggesting that sphingolipid metabolism may be associated with PV cell proliferation.

With the recent development of metabolomics technologies, an increasing number of metabolic biomarkers have been used to assess the prognosis of malignant tumors. Lo Presti et al. [43] developed an OPLS-DA model with two subgroups of favorable and unfavorable risk in AML patients and found higher levels of aspartate and glutathione in the bone marrow of AML patients in the unfavorable group than in AML patients in the favorable prognosis group, which was validated in subsequent follow-ups. In this study, Cer(d18:2/22:6-2OH(7S, 17S)), SM(d18:0/PGF1α), CerP(d18:1/16:0) and octadec-13-enoylcarnitine were selected as biomarkers of poor prognosis in PV patients by risk stratification and ROC curves. We also found that Cer(d18:2/22:6-2OH(7S, 17S)) and SM(d18:0/PGF1α) were positively correlated with both JAK2 mutational burden and WBC levels. One study reported that >50% of the JAK2 mutational burden is associated with fibrotic transformation [44,45], and leukocytosis is a risk factor for leukemic transformation and survival in PV patients [17,46]. This reinforces that they are risk factors for PV prognosis, suggesting that their effect on PV prognosis may be achieved through interaction with JAK2 mutational burden and stimulation of leukocyte proliferation, raising the risk of fibrosis or leukemic transformation. However, the prognostic biomarkers screened in this study remain to be further validated in the follow-up of patient survival.

However, there are still some limitations to this study. Firstly, organism metabolism is influenced by several factors, such as environment, genetics, and intestinal flora, which may affect the concentration of metabolites, and secondly, this study was subjected to a small sample size. Therefore, cellular or animal-level experiments and large-scale cohort studies are still needed for further validation.

## 5. Conclusions

Our study described the global metabolic profile of PV patients, analyzed the metabolites associated with JAK2 mutations and their relationship with blood cell counts, and screened metabolites associated with the prognosis of PV patients by risk stratification, providing new insights into the pathogenesis of PV and a reference for assessing patient prognosis. Notably, fatty acid metabolism, glucose metabolism, sphingolipid metabolism, and amino acid metabolism were significantly altered in PV. In addition, Cer(d18:2/22:6-2OH(7S, 17S)) and SM(d18:0/PGF1α) were closely associated with JAK2 mutations, which may contribute to the proliferation of peripheral blood cells in PV patients. The elevated levels of four potential biomarkers may provide a reference for poor prognosis in PV patients.

## Figures and Tables

**Figure 1 cancers-14-04913-f001:**
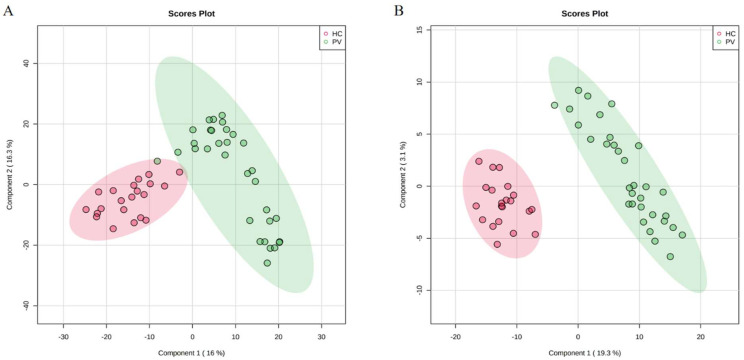
Score plots of serum metabolic profiles of the PV and HC groups. (**A**) PLS-DA score plots for the PV and HC groups in positive mode. (**B**) PLS-DA score plots for the PV and HC groups in negative mode.

**Figure 2 cancers-14-04913-f002:**
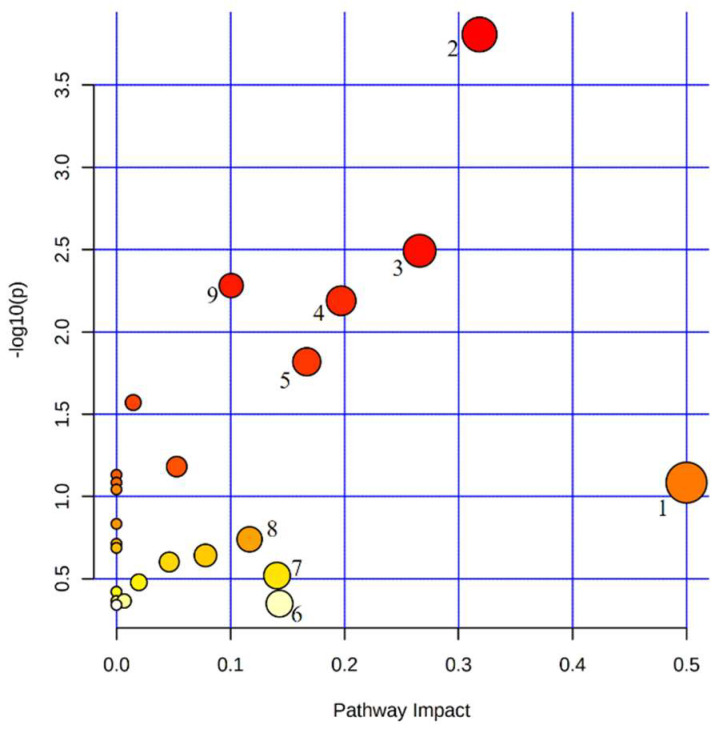
Metabolic pathways enrichment analysis of potential biomarkers in PV patients. 1. D-glutamine and D-glutamate metabolism. 2. Sphingolipid metabolism. 3. Pyruvate metabolism. 4. Alanine, aspartate and glutamate metabolism. 5. Arginine and proline metabolism. 6. Tryptophan metabolism. 7. Lysine degradation. 8. Arginine biosynthesis. 9. Glycolysis/gluconeogenesis.

**Figure 3 cancers-14-04913-f003:**
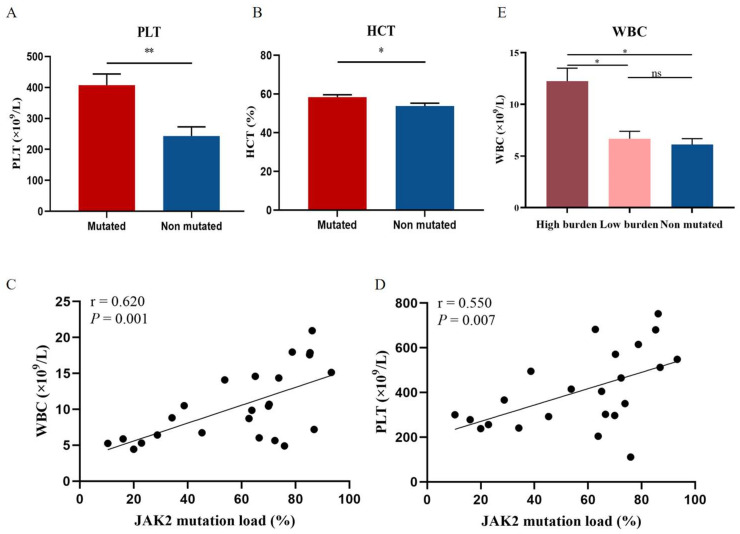
Correlation of JAK2 mutations and mutational burden with blood cell counts. Levels of PLT (**A**) and HCT (**B**) in JAK2-mutated patients compared with non-JAK2-mutated patients. (**E**) WBC levels in JAK2 high mutational burden, low burden, and unmutated groups. Mean value ± standard error of the mean (SEM). Mann–Whitney U test, * *p* < 0.05, ** *p* < 0.01, ns: no significance (*p* > 0.05). Scatter plot of the correlation between WBC (**C**) and PLT (**D**) levels and JAK2 mutational burden. Spearman’s correlation test.

**Figure 4 cancers-14-04913-f004:**
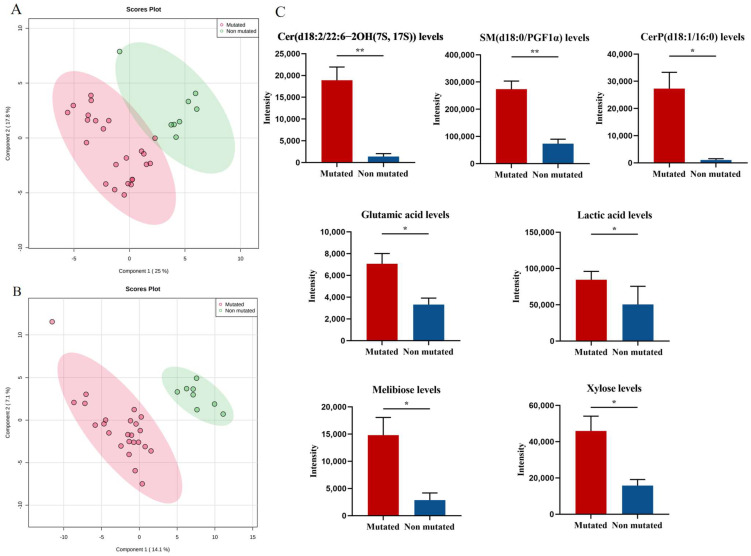
Specific metabolic profile of PV patients according to the mutational status of JAK2. (**A**) PLS-DA score plots for the JAK2-mutated group and non-JAK2-mutated group in positive ion mode. (**B**) PLS-DA score plots for the JAK2-mutated group and non-JAK2-mutated group in negative ion mode. (**C**) Levels of Cer(d18:2/22:6-2OH(7S, 17S)), SM(d18:0/PGF1α), CerP(d18:1/16:0), glutamic acid, lactic acid, melibiose and xylose in JAK2-mutated patients compared with non-JAK2-mutated patients. Mean value ± SEM. Mann–Whitney U test, * *p* < 0.05, ** *p* < 0.01.

**Figure 5 cancers-14-04913-f005:**
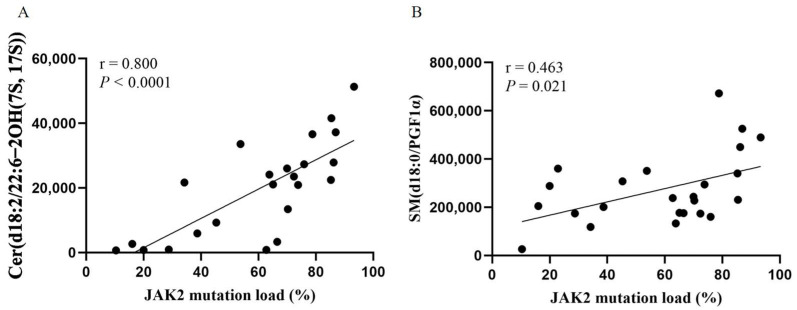
Scatter plot of the correlation between Cer(d18:2/22:6-2OH(7S, 17S)) (**A**) and SM(d18:0/PGF1α) (**B**) levels and JAK2 mutational burden. Spearman’s correlation test.

**Figure 6 cancers-14-04913-f006:**
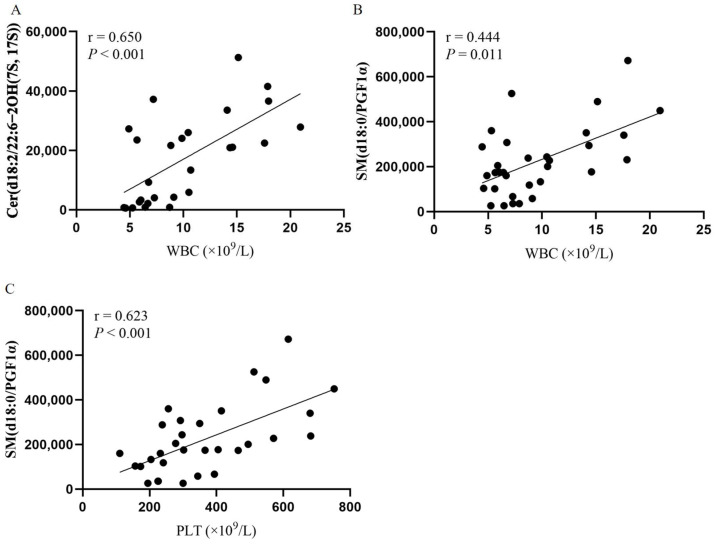
Correlation of Cer(d18:2/22:6-2OH(7S, 17S)) and SM(d18:0/PGF1α) with blood cell counts. Spearman’s correlation test. (**A**) Scatter plot of the correlation between Cer(d18:2/22:6-2OH(7S, 17S)) and WBC levels. (**B**) Scatter plot of the correlation between SM(d18:0/PGF1α) and WBC levels. (**C**) Scatter plot of the correlation between SM(d18:0/PGF1α) and PLT levels.

**Figure 7 cancers-14-04913-f007:**
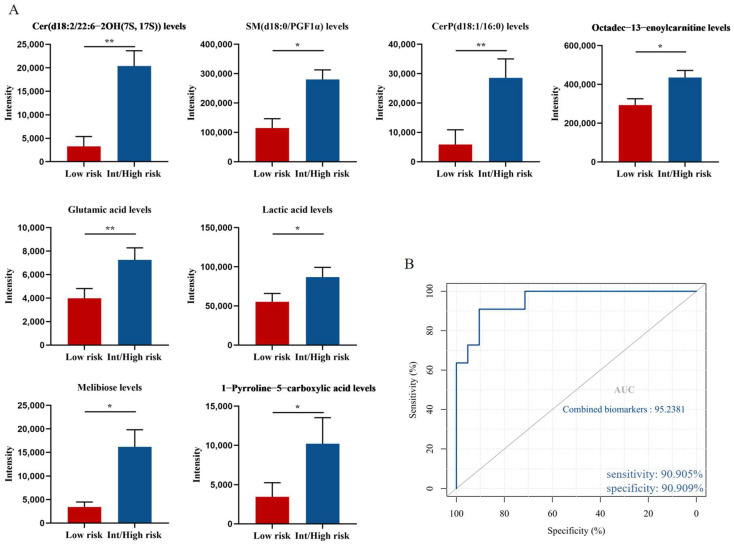
Potential biomarkers related to the prognosis of PV patients. (**A**) Levels of Cer(d18:2/22:6-2OH(7S, 17S)), SM(d18:0/PGF1α), CerP(d18:1/16:0), octadec-13-enoylcarnitine, glutamic acid, lactic acid, melibiose and 1-pyrroline-5-carboxylic acid in intermediate/high-risk patients compared with low-risk patients. Mean value ± SEM. Mann–Whitney U test, * *p* < 0.05, ** *p* < 0.01. (**B**) ROC curve analysis for the combination of 4 biomarkers.

**Table 1 cancers-14-04913-t001:** Potential biomarkers for PV patients.

No.	Metabolite ^a^	Mass	Retention Time/Min	Tendency ^b^	Ion Mode	Pathway
1	Arachidic acid	312.3028	0.9840	Up	Positive	Fatty acid metabolism
2	Dodecanoic acid	200.1776	1.4151	Up	Positive	Fatty acid metabolism
3	Myristic acid	228.2089	1.3200	Up	Positive	Fatty acid metabolism
4	Palmitic acid	256.2402	1.4951	Up	Positive	Fatty acid metabolism
5	Octadec-13-enoylcarnitine	425.3505	0.9802	Up	Positive	Fatty acid oxidation
6	Dodec-9-enedioylcarnitine	371.2308	1.3390	Up	Positive	Fatty acid oxidation
7	Dodecanoylcarnitine	344.2795	1.3580	Up	Positive	Fatty acid oxidation
8	Hexacosanoyl carnitine	539.4914	0.9802	Up	Positive	Fatty acid oxidation
9	Octanoylcarnitine	288.2175	1.4684	Up	Positive	Fatty acid oxidation
10	Glucose	180.0634	5.0102	Up	Negative	Glycolysis/Gluconeogenesis
11	Lactic acid	90.0317	4.2093	Up	Negative	Glycolysis/Gluconeogenesis
12	Pyruvic acid	88.0160	3.4841	Down	Negative	Glycolysis/Gluconeogenesis
13	Melibiose	342.1162	9.6866	Up	Positive	Galactose metabolism
14	Xylose	150.0528	3.0263	Up	Negative	Pentose and glucuronate interconversions
15	Cer(d18:1/24:1)	647.6216	0.8812	Up	Positive	Sphingolipid metabolism
16	Cer(d18:2/6 keto-PGF1α)	649.4918	0.8051	Up	Positive	Sphingolipid metabolism
17	CerP(d18:1/16:0)	617.4784	4.1358	Up	Positive	Sphingolipid metabolism
18	Phytosphingosine	317.2930	0.9840	Up	Positive	Sphingolipid metabolism
19	SM(d18:0/12:0)	650.5363	4.7082	Up	Positive	Sphingolipid metabolism
20	SM(d18:0/22:0)	788.6771	4.4175	Down	Positive	Sphingolipid metabolism
21	SM(d18:0/PGF1α)	804.5993	0.9878	Up	Positive	Sphingolipid metabolism
22	Sphingosine	299.2824	1.3542	Up	Positive	Sphingolipid metabolism
23	Cer(d18:2/22:6-2OH(7S, 17S))	639.4863	4.1180	Up	Negative	Sphingolipid metabolism
24	SM(d20:1/PGE2)	826.5836	2.8587	Up	Negative	Sphingolipid metabolism
25	SM(d20:1/PGF2α)	828.5993	2.6873	Up	Negative	Sphingolipid metabolism
26	Glutamic acid	147.0532	2.5652	Up	Negative	Glutamate metabolism
27	1-Pyrroline-5-carboxylic acid	113.0477	3.2366	Up	Negative	Glutamate Metabolism
28	Lactoylglutathione	379.1049	1.1106	Down	Positive	Pyruvate metabolism
29	Aminoadipic acid	161.0688	0.8012	Up	Positive	Lysine biosynthesis
30	Homo-L-arginine	188.1273	10.8460	Down	Positive	Arginine metabolism
31	Tryptophan	204.0899	1.6445	Up	Positive	Tryptophan metabolism
32	Cytidine	243.0855	3.2290	Up	Negative	Pyrimidine metabolism
33	Bilirubin	584.2635	0.8703	Up	Negative	Porphyrin metabolism

^a^ Metabolites were identified using available library databases and standard samples. ^b^ Tendency in the PV group compared to the HC group.

## Data Availability

The datasets generated during and/or analyzed during the current study are available from the corresponding author on reasonable request.

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
