# Peer review of "Metabolic Biomarkers Affecting Cell Proliferation and Prognosis in Polycythemia Vera"

_cancers, 2022, doi:10.3390/cancers14194913_

Round 1

Reviewer 1 Report

In this study, the authors analyzed metabolic differences in peripheral blood sera between 32 Polycythemia Vera (PV) patients 21-77 years of age with a median of 54.5 years, below the 65 years of median age of PV onset, and 20 healthy controls (HC), 28-78 years of age with a median of 49.5 years, by liquid chromatography-mass spectrometry (LC-MS).

  The goal is to analyze differences in endogenous metabolic profile between PV and HC samples and their relationship with JAK2 mutations, occurring in 80-90% of PV patients, and blood cell proliferation and screen for prognosis-related metabolic biomarkers. 

Results

Score plots of partial least squares discriminant analysis (PLS-DA) showed metabolic changes occurring in the 32 PV patients’ group compared to the 20 HC group, in both positive-ion and negative-ion chromatography. The authors identified 33 endogenous metabolites with significant changes in their serum levels in PV compared to HC samples (Variable Importance in Projection score >1; Fold Change >2; t-test P<0.05). The potential biomarkers are identified prevalently as fatty acids, sphingolipids, acylcarnitine, sugars, and amino acids. Metabolic pathway analysis identified 9 pathways affected by metabolic changes in PV, which can be attributed to the metabolism of fatty acids, sphingolipids, glucose, and amino acids. 

The effect of JAK2 mutations and dysregulated activation of the JAK-STAT pathway on white blood cells (WBC), platelet (PLT), and hematocrit (HCT) was measured by cell count in JAK2-mutant (n=24) compared to non-JAK2-mutant (n=8) patients. PLT and HCT levels were found significantly higher in JAK2-mutant PV patients, but not WBC levels. In JAK2-mutant PV patients, a significant positive correlation was found between JAK2 mutation load and WBC and PLT count levels.

PLS-DA score plots revealed separation in metabolic profiles between JAK2-mutant and non-JAK2-mutant patient groups, and 7 of the 33 endogenous metabolites identified as potential PV biomarkers revealed higher levels in JAK2-mutant compared with the non-JAK2-mutant PV patient group. They comprise 3 sphingolipids, 1 amino acid, 1 organic acid, and 2 sugars. Five of these 7 metabolites show significant Spearman’s correlation with WBC count, two of which revealed positive correlations of their levels with JAK2 mutation load and WBC count in one, and with JAK2 mutation load, WBC, and PLT count in the other.

Eight of the 33 endogenous metabolites identified as potential PV biomarkers showed significant elevation in an intermediate/high-risk group of PV patients (n=21) when compared to a low-risk group of PV patients (n=11) stratified according to the prognostic model developed by Tefferi et al [Ref 17]. According to binary logistic regression and receiver operating characteristic (ROC) curve analyses, 4 of these intermediate/high-risk PV group metabolic biomarkers could be used as potential metabolic biomarkers of poor prognosis in patients with PV. 

Major points

In Table 1, the Tendency of many endogenous metabolites (e.g., Lactic Acid, SM(d18:0/PGF1α), Glutamic acid, Cytidine, Hexacosanoyl carnitine, Octadec-13-enoylcarnitine) identified as potential biomarkers in the PV group compared to the HC group does not match with the visual heatmap results in Suppl Figure 1. Particularly, despite higher levels in JAK2-mutant compared to non-JAK2-mutant PV patients in Figure 4C, SM(d18:0/PGF1α) metabolite shows overall PV group lower heatmap levels than in the HC group. The authors should explain this discrepancy. JAK2 mutation status should be indicated on the Hierarchical clustering heatmap in Suppl Figure 1. 

In Figure 3, how do the authors explain a WBC count positive correlation with JAK2 mutation load in the absence of a significant WBC count difference between JAK2-mutant and non-JAK2-mutant PV patients?

The type of JAK2 mutation (e.g., JAK2 exon 12 mutations when JAK2V617F is not detected) in PV patients, their method of detection and analysis, and the method of calculation of JAK2 mutation load should be indicated in Material and Methods and Figures, and the results should be summarized in a Table. 

Six of the 7 metabolites identified in the JAK2-mutant versus non-JAK2-mutant PV patients in Figure 4 are shared among the 8 metabolites identified in the intermediate/high-risk versus low-risk PV patient group in Figure 7. Of the 4 combined biomarkers of poor prognosis identified in the intermediate/high-risk PV patient group by ROC curves, 3 are shared with the JAK2-mutant PV patients and a fourth, Octadec-13-enoylcarnitine, shows overall decreased levels in PV compared to HC samples in hierarchical clustering heatmap.

The paper focuses on the identification of JAK2 mutation-associated metabolic profiles and their correlation with increased blood cell proliferation due to JAK-STAT pathway dysregulation in PV patients. The scope is to better understand the pathogenesis associated with PV metabolism, its intrinsic connection with JAK2 mutations, cell proliferation, and prognosis. Authors should better emphasize this by running multiple logistic regression and ROC curve analysis to identify combined metabolic biomarkers of poor prognosis in PV patients carrying JAK2 mutations, one of the main diagnostic criteria for PV, to distinguish them from combined metabolic biomarkers of poor prognosis in overall PV.

For instance, what are the AUC value, sensitivity, and specificity of Cer(d18:2/22:6-2OH(7S, 17S)), CerP(d18:1/16:0) and/or others as poor prognosis combined metabolic biomarkers in JAK2-mutant PV patients? 

Minor points

In result session 3.1 the sentence: “Ten chromatographic peaks were randomly selected and the RSD values of peak area and retention time were calculated”, is duplicated.

Author Response

Thanks for your comments on our manuscript. We have revised our paper according to your comments. Please see the attachment.

Reviewer 2 Report

The authors present a descriptive assessment of metabolomics in patients with polcythemia and seek to tie metabolic profiles with prognosis/pathogenesis. This provides initial insight into a potential future therapeutic target.

Major

Methods

1. Healthy samples were collected from volunteers without metabolic disorders. What specifically do the authors mean by this and how was it verified?

2. Eight of the P vera patients did not have a JAK2 mutation. This is 8/32 patients or 25% which is outside of what one would expect in a regular P vera population. Were these JAK2 exon12 mutations? Why do the authors believe this breakdown in patients occurred or was it intentionally selected? If selected intentionally then please provide a clear rationale for why especially consider the small sample size of the study. This would then need to be included in the limitations section. 

3. A description of the prognostic model for P vera (Reference 17) should be included in the methods. Further I assume the prognostic model was applied at initial diagnosis but would clarify this in the methods. 

Minor Comments

Introduction

1. Based on SEER database results the incidence of P vera is 1 per 100,000 not 1 per 10 million. 

2. Page 2 "With the maturation of research mechanisms...." This sentence does not have an ending. 

Methods

1. IRB approval should be mentioned in the firs sentence of the methods section. 

Author Response

(The authors gave the same response as above.)

Reviewer 3 Report

The manuscript entitled “Metabolic Biomarkers Affecting Cell Proliferation and Prognosis in Polycythemia Vera” analyzed the metabolic differences between PV patients and healthy controls by LC-MS and correlated them with cell proliferation and prognosis, identifying a metabolic biomarker of high-risk patients. This manuscript is well-written and designed. In my opinion, the major limitations of this manuscript are the limited number of studied patients and the low follow-up time (do not allow survival analysis). However, the Authors need to address some minor issues.

1. The patient’s clinical and demographic data should be improved and placed in the main text. The Authors should include data stratified according to the absence and presence of JAK2 mutations and the demographic data of healthy donors (groups compared in the manuscript). They should also include the statistical analysis of these groups to ensure that metabolic differences observed in the manuscript have no bias (or, if needed, discuss the bias).

2. In the Material and Methods section, precise centrifugation instructions should be included. So, the Authors should replace RPMs with g (page 3) since RPMs differ with rotor size and g force will stay the same.

3. Figure 1 is not cited in the text.

4. Figures 4A and 4B are too small and hard to read.

5. On page 10, the Authors said “A previous study found …”, but no citation is indicated.

6. In Figure 7B, the Authors should include the sensitivity and specificity values.

Author Response

(The authors gave the same response as above.)

Round 2

Reviewer 1 Report

In Material and Methods, the newly added description of the calculation method of JAK2 mutation load can be improved. For instance, what do the authors mean by effective coverage area (panel interval)?

Author Response

(The authors gave the same response as above.)
